# Molecular and epidemiological analysis of a *Burkholderia cepacia* sepsis outbreak from a tertiary care hospital in Bangladesh

**Refath Farzana**[1][☉]*, **Lim S. Jones**[2], **Md. Anisur Rahman**[3], **Kirsty Sands**[1], **Edward Portal**[1], **Ian Boostrom**[1], **Md. Abul Kalam**[4], **Brekhna Hasan**[1], **Afifah Khan**[1], **Timothy R. Walsh**[1][☉]

**1** Department of Medical Microbiology, Institute of Infection and Immunity, School of Medicine, Cardiff University, Cardiff, United Kingdom, **2** Public Health Wales Microbiology, University Hospital of Wales, Cardiff, United Kingdom, **3** Department of Virology, Dhaka Medical College, Dhaka, Bangladesh, **4** Department of Burn and Plastic Surgery, Sheikh Hasina National Institute of Burn and Plastic Surgery, Dhaka, Bangladesh

☉ These authors contributed equally to this work.
* FarzanaR@cardiff.ac.uk

**Data Availability Statement:** All relevant data are within the manuscript and its Supporting

## Abstract

### Background

*Burkholderia cepacia* complex (Bcc) is a group of serious pathogens in cystic fibrosis patients and causes life threatening infections in immunocompromised patients. Species within the Bcc are widely distributed within the environment, can survive in the presence of disinfectants and antiseptics, and are inherently multidrug resistant (MDR).

### Methods

Dhaka Medical College Hospital (DMCH) patients with a *B. cepacia* positive blood culture between 20 October 2016 to 23rd September 2017 were considered as outbreak cases. Blood stream infections (BSIs) were detected using BacT/ALERT 3D at DMCH. *B. cepacia* was isolated on chromogenic UTI media followed by MALDI-TOF. Minimum inhibitory concentration (MIC) of clinically relevant antibiotics was determined by agar dilution. Whole genome sequencing was performed on an Illumina MiSeq platform. Patients' demographic and clinical data were collected. Patients' clinical history and genomic data of the outbreak strains were merged to investigate possible outbreaks. Ninety-one *B. cepacia* genomes were downloaded from '*Burkholderia* Genome Database' and the genomic background of the global strains were compared with our outbreak strains.

### Results

Among 236 BSIs, 6.35% (15/236) were *B. cepacia*. Outbreak cases were confined to the burn critical care unit and, to a lesser extent, the paediatrics department. There was a continuum of overlapping cases at DMCH between 23 October 2016 to 30 August 2017. Core genome SNPs showed that the outbreak strains were confined to a single clade, corresponded to a common clone (ST1578). The strains were shown to be MDR and associated with a mortality of 31% excluding discharge against medical advice. MIC profiles of the

Information files. Accession numbers have been mentioned in the manuscript.

**Funding:** The study was supported by Commonwealth Scholarship Commission. Refath Farzana is the recipient of the Commonwealth Fellowship (BDCS-2016-53). The funders had no role in study design, data collection and analysis, decision to publish, or preparation of the manuscript.

**Competing interests:** The authors have declared that no competing interests exist.

strains suggested that antibiotics deployed as empirical therapy were invariably inappropriate. The genetic background of the outbreak strains was very similar; however, a few variations were found regarding the presence of virulence genes. Compared to global strains from the *Burkholderia* Genome Database, the Bangladeshi strains were genetically distinct.

## Conclusions

Environmental surveillance is required to investigate the aetiology and mode of transmission of the *B. cepacia* outbreak. Systematic management of nosocomial outbreaks, particularly in resource limited regions, will mitigate transmission and will improve patients' outcomes.

### Author summary

The Governmental health system in Bangladesh is free to the general public but given the huge burden (4–5 times hospital capacity), the Governmental hospitals are always overcrowded, and infection control is minimal. Antibiotics are used empirically to manage infections and invariably offered to all patients admitted. Outbreaks are a regular phenomenon in the public hospitals of Bangladesh which are rarely fully analysed. Herein, we report a *B. cepacia* outbreak from a burn unit for a protracted period where the patients were treated with inappropriate antibiotic therapies. *B. cepacia* is a Gram-negative bacterium, mostly a lung pathogen in cystic fibrosis patients but can also produce infections in immunosuppressed patients. The epidemiology and molecular data from the outbreak strains indicate the need for interventions and improved infection control programs to manage outbreaks in Bangladeshi hospitals.

## Introduction

The genus *Burkholderia* incorporates Gram negative, catalase-producing, lactose-nonfermenting), obligately aerobic bacilli and encompasses the species of the *Burkholderia cepacia complex* (Bcc), *Burkholderia mallei*, *B. pseudomallei*. and *B. gladioli* [1]. Bcc is composed of at least 20 different species, including *B. cepacia*, *B. multivorans* and *B. cenocepacia* [1,2]. Bcc was first described in mid-1980s, in cystic fibrosis (CF) patients as a cause of 'Cepacia Syndrome', characterized by lung function deterioration, bacteremia and death [3]. The members of Bcc are widely distributed in the environment, including water, soil, fruits and vegetables and can survive for prolong periods of time in moist environments, even in the presence of disinfectants and antiseptics. They are capable of colonizing fluids in the hospital such as irrigation solutions or intravenous fluids and, serve as potential source of nosocomial infections [4,5,6]. Bcc rarely cause diseases in healthy individuals and is mostly regarded as a serious pathogen in cystic fibrosis patients [7]. The pathogens are inherently multidrug resistant (MDR) and highly transmissible by direct contact [7–9].

Regardless of geographical location nosocomial outbreaks attributed by Bcc are frequent. Recent outbreaks of Bcc bacteraemia from India, Germany and USA, have been reported particularly in the immunocompromised patients [6, 10–12]. Bcc outbreaks are always linked to hospital environment and/or contaminated medical devices, intravenous fluids or antiseptic solutions [4–6,10–13]. We describe an outbreak of *B. cepacia* bacteraemia in burn's critical care units of Dhaka Medical College Hospital (DMCH) analysed by whole genome

sequencing. The population structure and genetic relationship of *B. cepacia* isolated in this study were compared with 91 global *B. cepacia* deposited previously in the '*Burkholderia* Genome Database' [14].

## Methods

### Ethical statement

Ethical consent was given by the Ethical Review Committee (ERC) of Dhaka Medical College Hospital (DMCH) prior to the start of the study and in accordance with the Helsinki Declaration [Memo no: MEU-DMC/ECC/2017/122] [15]. Required written consent was taken from all the participants for this study.

### Hospital setting and outbreak case definition

We carried out this study between 20 October 2016 and 23rd September 2017 in DMCH which is the largest public hospital in Bangladesh. A total of 527 blood culture samples received by the clinical microbiology laboratory were included in this study. Detailed microbiological analysis of isolates, including Whole Genome Sequencing (WGS), performed at Cardiff University, were undertaken retrospectively. Patients with a *B. cepacia* positive blood culture were included as outbreak cases. Patients' demographic (name, age, sex, locality and socio-economic condition) and clinical data (clinical symptoms or reason for hospitalization, ward, type of infection, admission date, sample collection date, outcome and antimicrobial therapy) were collected. Clinical data was collected following positive culture results from the local laboratory. Additional clinical follow up information was not available in the current study, except for outcome (discharge, death, discharge against medical advice {DAMA}). The study outline is described in Fig 1.

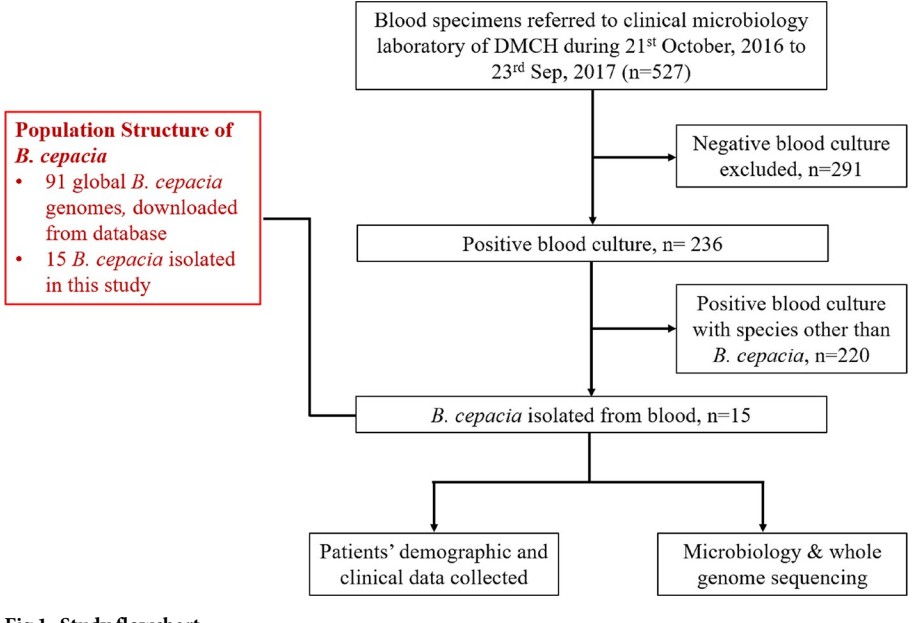

**Fig 1. Study flowchart.**

## Purification and screening of phenotypic resistance to the isolates of interest

Blood stream infections were detected using BacT/ALERT 3D (bioMerieux, North Carolina, USA) at DMCH. The isolates were sub-cultured onto chromogenic UTI agar (Liofilchem, Roseto, Italy) and the species were identified by Matrix-Assisted Laser Desorption/Ionization-Time of Flight (MALDI-TOF) mass spectrometry (MS) (Bruker Daltonics, Bremen, Germany). Minimum inhibitory concentrations (MIC) to clinically relevant antimicrobials (amoxicillin-clavulanate, piperacillin-tazobactam, ceftriaxone, ceftazidime, cefotaxime, cefepime, imipenem, meropenem, ciprofloxacin, levofloxacin, amikacin, gentamicin, sulfamethoxazole-trimethoprim, fosfomycin, tigecycline and colistin) were determined by agar dilution and interpreted according to Clinical and Laboratory Standards Institute (CLSI) breakpoints [16]. MIC determination was carried out in triplicate.

## Illumina MiSeq sequencing

WGS was performed on the Illumina MiSeq platform (Illumina Inc., San Diego, CA). DNA libraries were prepared for paired end sequencing (2x301 cycles) using Nextera XT v2. Genomic DNA was extracted from overnight culture using the QIAcube (Qiagen, Hilden, Germany), and resulting gDNA was quantified using the Qubit 3.0 (Thermos Fisher Scientific, Waltham, USA). Quality control of raw reads included fastqc (v0.11.2) and adaptor trimming was performed using Trimgalore (v0.4.3). Reads were assembled into contigs using the *de novo* assembler SPAdes (v3.9.0) (.fasta) and were aligned to the original fastq reads using Burrows-Wheeler aligner (BWA) (v0.7.15). Any error was corrected using Pilon (v1.2). Assembly metrics were evaluated using Quast (v2.1). The *de novo* assembly produced multiple contigs (145–330), with a mean GC content of 50% and mean coverage 12X, annotated with Prokka (v1.12). Resistance genes were retrieved from Comprehensive Antibiotic Resistance Database (CARD) [17]. Mutations in *ampD* were mapped to *ampD* (BCAL3430) of *B. cenocepacia* strain J2315 which possesses a ceftazidime MIC of 2 µg/ml to evaluate the putative cause of ß-lactam resistance [14,18]. Virulence factors (VFs) were evaluated by Virulence Factor Database (VFDB) [19].

## Multi locus sequence typing (MLST)

*B. cepacia* (n = 15) isolated in this study was submitted to pubMLST to assign as sequence types (STs) based on 7 loci [20]. Global *B. cepacia* (n = 91) genomes were downloaded from '*Burkholderia* Genome Database' of which 82 strains could not be matched with previously described STs [14]. We also submitted the 82 genomes from database to pubMLST [20]. A minimum spanning phylogenetic tree was constructed with 106 *B. cepacia* genome using Ridom SeqSphere+ (Ridom GmbH, Münster, Germany) based on 7 *B. cepacia* MLST genes.

## Core genome phylogenetic analysis

We mapped single nucleotide polymorphisms (SNPs) of global *B. cepacia* from database (n = 91) and our outbreak strains (n = 15) with the first isolated strain in our study, dm93 (used as reference genome) using Snippy (3.2) [21]. SNPs alignemnt was performed to build a high-resolution phylogeny. We constructed a maximum likelihood (ML) tree using FastTree (2.1.3). We employed Interactive Tree of Life (iTOL) to visualize the tree [22].

### Investigating possible outbreaks

Data extracted from patients' clinical history and WGS data of the outbreak strains were evaluated to investigate possible outbreaks. Likely epidemiological links on transmission (patient to patient connectivity) was predicted if there is overlapping of hospital stay among the outbreak cases (Fig 2). Detailed epidemiological information and subsequent infection control measures were not available as the microbiological information was only available retrospectively. Infection Prevention and Control (IPC) measures were not implemented at the time due to the limited and untimely microbiological information at the DMCH.

### Accession numbers

WGS data of all *B. cepacia* in this study were deposited in the National Centre for Biotechnology Information under accession numbers stated below.

| Accession | Organism |
|---|---|
| SNSF00000000 | *Burkholderia cepacia* b219 |
| SNSG00000000 | *Burkholderia cepacia* b212 |
| SNSH00000000 | *Burkholderia cepacia* b163 |
| SNSI00000000 | *Burkholderia cepacia* b124 |
| SNSJ00000000 | *Burkholderia cepacia* b111 |
| SNSK00000000 | *Burkholderia cepacia* b101 |
| SNSL00000000 | *Burkholderia cepacia* b100 |
| SNSM00000000 | *Burkholderia cepacia* b98 |
| SNSN00000000 | *Burkholderia cepacia* b84 |
| SNSO00000000 | *Burkholderia cepacia* b64 |
| SNSP00000000 | *Burkholderia cepacia* b19 |
| SNSQ00000000 | *Burkholderia cepacia* b13 |
| SNSR00000000 | *Burkholderia cepacia* b6 |
| SNSS00000000 | *Burkholderia cepacia* dm93 |
| SSHL00000000 | *Burkholderia cepacia* b99 |

## Results

### Study population and overview of *B. cepacia* cases

Among 527 blood culture specimens, 45% (236/527) were culture positive and 3% (15/527) were *B. cepacia* bacteraemia (Fig 1). Of the 15 *B. cepacia* infected patients, 7 (46.7%) were male and 8 (53.3%) were female; the mean (±SD) age was 14 years (±12.57). All the patients in this cohort belonged to a low socio-economic group (lower middle to below the national poverty level). The mean (±SD) hospital stay of *B. cepacia* cases was 43.9 days (±27.1). Antibiotic usage among the cases with *B. cepacia* septicaemia was: ceftriaxone, 80.0%; amikacin, 53.3%; levofloxacin, 33.3%; ceftazidime, 20%; meropenem, 13.3%, colistin, 13.3%; flucloxacillin, 6.7%; azithromycin, 6.7%; gentamicin, 6.7%; clindamycin, 6.7%; cefuroxime, 6.7% (Table 1). Two of the *B. cepacia* cases took discharge against medical advice (DAMA). Not including DAMA, the mortality rate was 31% (4/13) (Fig 2; Table 1). Patient characteristics are illustrated in Table 1. The first case was admitted to paediatrics department of DMCH on the 23rd of October 2016. According to hospital records, the patient was a suspected case of sepsis, and blood was sent for culture and sensitivity on the 2nd of November 2016 which was positive for *B. cepacia*. The rest of the cases were confined to the burn HDU or the burn paediatric HDU or the burn ICU (Fig 2; Table 1). Excluding DAMA (5/135), in the burn unit, the mortality rate for patients

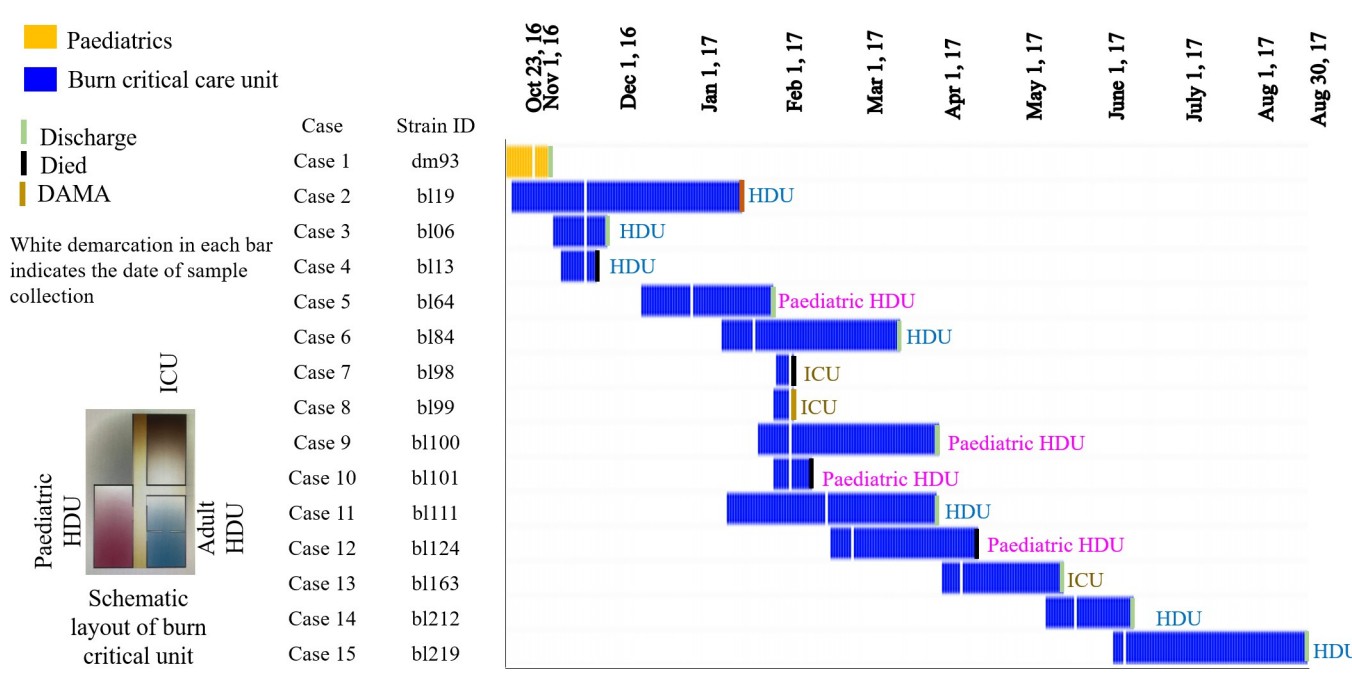

**Fig 2. Length of hospital stay of the patients with *B. cepacia* bacteraemia.**

**Table 1. Clinical characteristics of patients infected with *B. cepacia*.**

| Strain ID | Cases | Ward | Sample | Age[a] | Sex | SEC | DOA | DOSC | DOD | THS | Underlying disease | Antibiotic history | Outcome |
|---|---|---|---|---|---|---|---|---|---|---|---|---|---|
| dm93 | Case 1 | Pae | Blood | 2.5 | M | BPL | 23.10.16 | 02.11.16 | 7.11.16 | 16 | Unknown[b] | CRO, GEN | Discharge |
| b19 | Case 2 | Burn HDU | Blood | 30 | F | BPL | 25.10.16 | 22.11.16 | 21.1.17 | 58 | 35% FB | CRO | DAMA |
| b06 | Case 3 | Burn HDU | Blood | 22 | M | Poor | 10.11.16 | 22.11.16 | 30.11.16 | 21 | 35% FB | CFU, CLN | Discharge |
| b13 | Case 4 | Burn HDU | Blood | 45 | F | BPL | 13.11.16 | 22.11.16 | 26.11.16 | 14 | 25% FB | AK, CAZ | Died |
| b64 | Case 5 | Burn Pae. HDU | Blood | 3.5 | F | Poor | 14.12.16 | 02.01.17 | 2.2.17 | 51 | 15% FB | AZ, CL, CRO | Discharge |
| b84 | Case 6 | Burn HDU | Blood | 15 | M | BPL | 14.01.17 | 26.01.17 | 23.3.17 | 69 | 40% EB | AK, CAZ, MEM | Discharge |
| b98 | Case 7 | Burn ICU | Blood | 5 | F | BPL | 04.02.17 | 09.02.17 | 10.2.17 | 7 | 45% FB with II | AK, CRO, FLU | Died |
| b99 | Case 8 | Burn ICU | Blood | 10 | F | LM | 03.02.17 | 09.02.17 | 09.02.17 | 7 | 30% MB[c] | CRO, LEVO | DAMA |
| b100 | Case 9 | Burn Pae. HDU | Blood | 2.5 | F | LM | 28.01.17 | 09.02.17 | 8.4.17 | 71 | 22% SB | AK, CRO, LEVO | Discharge |
| b101 | Case 10 | Burn Pae. HDU | Blood | 7 | M | BPL | 03.02.17 | 09.02.17 | 17.2.17 | 15 | 43% FB | AK, CRO | Died |
| b111 | Case 11 | Burn HDU | Blood | 19 | M | Poor | 16.01.17 | 23.02.17 | 7.4.17 | 82 | 30% FB | AK, CRO, MEM | Discharge |
| b124 | Case 12 | Burn Pae. HDU | Blood | 3.5 | F | BPL | 25.02.17 | 05.03.17 | 23.4.17 | 58 | 30% FB | CAZ, CL, CRO | Died |
| b163 | Case 13 | Burn ICU | Blood | 1.5 | M | Poor | 10.4.17 | 17.4.17 | 26.5.17 | 46 | 40% SB | CRO, LEVO | Discharge |
| b212 | Case 14 | Burn HDU | Blood | 23 | F | Poor | 20.5.17 | 31.5.17 | 22.6.17 | 34 | 35% FB | AK, CRO, LEVO | Discharge |
| b219 | Case 15 | Burn HDU | Blood | 20 | M | Poor | 15.6.17 | 19.6.17 | 30.8.17 | 76 | 35% EB with II | AK, CRO, LEVO | Discharge |

Cases were delineated according to date of isolation of *B. cepacia* chronologically. ICU, intensive care unit; HDU, high dependency unit, M, male; F, female; SEC, socio-economic condition; BPL, below poverty level, LM, lower middle; DOA, date of admission; DOSC, date of sample collection; DOD, date of discharge/DAMA/death; THS, total hospital stay; Pae, paediatrics; FB, flame burn, EB, electric burn, II, inhalation injury; MB, mixed bun; SC, scald burn; DAMA, discharge against medical advice. AK, amikacin; AZ, azithromycin; CAZ, ceftazidime; CFU, cefuroxime; CL, colistin; CLN, clindamycin; CRO, ceftriaxone; FLU, flucloxacillin; GEN, gentamicin; LEVO, levofloxacin; MEM, meropenem

[a]Age are given in years

[b]Underlying disease was not available in hospital record; however, this case was clinically suspected as sepsis and therefore, blood was referred for culture

[c]Combination of chemical and flame burn

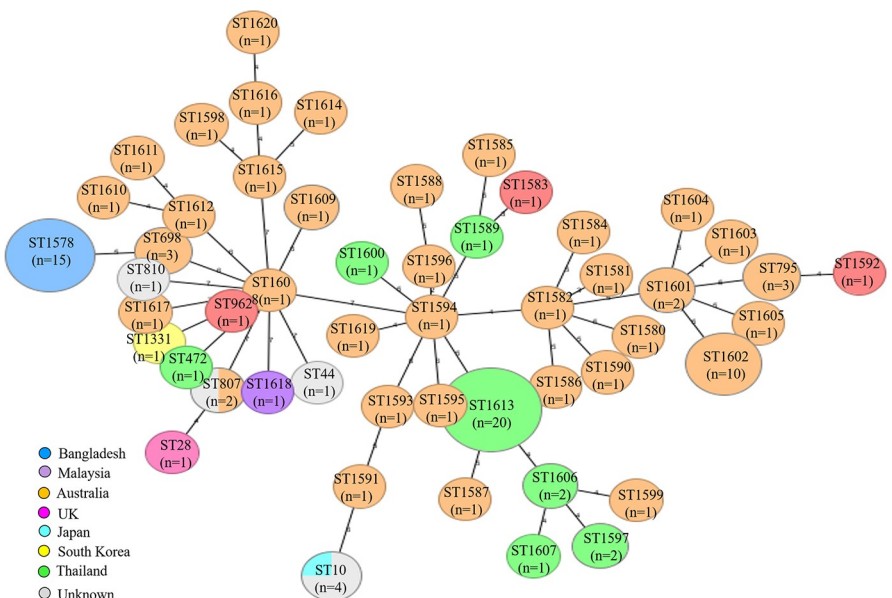

**Fig 3. Minimum spanning tree of *B. cepacia* by MLST type.** Each node within the tree represents a single ST. The size of the nodes is proportional to the number of isolates represented by corresponding node. Selected nodes are labelled with corresponding STs, and number of isolates represented. All global strains including Bangladeshi outbreak strains mentioned in this diagram were assigned as corresponding STs in this study except ST10, ST44, ST807 and ST810.

with other bacterial sepsis was higher (60.17%, 71/118) compared to *B. cepacia* sepsis (33.33%, 4/12) ($p<0.1$).

## Clonal relationship with global *B. cepacia*

*B. cepacia* isolated in this study were assigned as a single novel ST, ST1578 (Fig 3). All strains (n = 106) were grouped into 9 clusters (Fig 3). The Bangladeshi strains shared ST clusters with strains from Australia, USA, South Korea, Thailand, UK and Malaysia. Interestingly, STs differed according to geographical area (Fig 3). Core genome alignment also suggested that the Bangladeshi *B. cepacia* are genetically closer to environmental strains of Australian origin rather than human strains isolated from cystic fibrosis patients in the UK (LMG16656.fsa nt), Thailand (LO6.fasta), and other Asian strains (Fig 4).

## Outbreak investigation

We documented this outbreak between October 2016 and August 2017 at DMCH (Fig 2). All outbreak cases had clinical signs of sepsis at the time of sample collection and patients in burn critical units were undergoing artificial ventilation and had central venous catheter lines and urinary catheters at the time of diagnosis. There was a continuous overlapping of patients at DMCH during the outbreak period (Table 1; Fig 2). Core genome SNPs showed that the outbreak strains were confined to a single clade (Fig 4; S1 Data), corresponded to a common clone (ST1578) (Fig 3).

## Background on resistance and virulence

MICs were performed in triplicate and all repetitions were within one-dilution. There are no MIC breakpoints for *B. cepacia* recommended by EUCAST [23]. According to CLSI

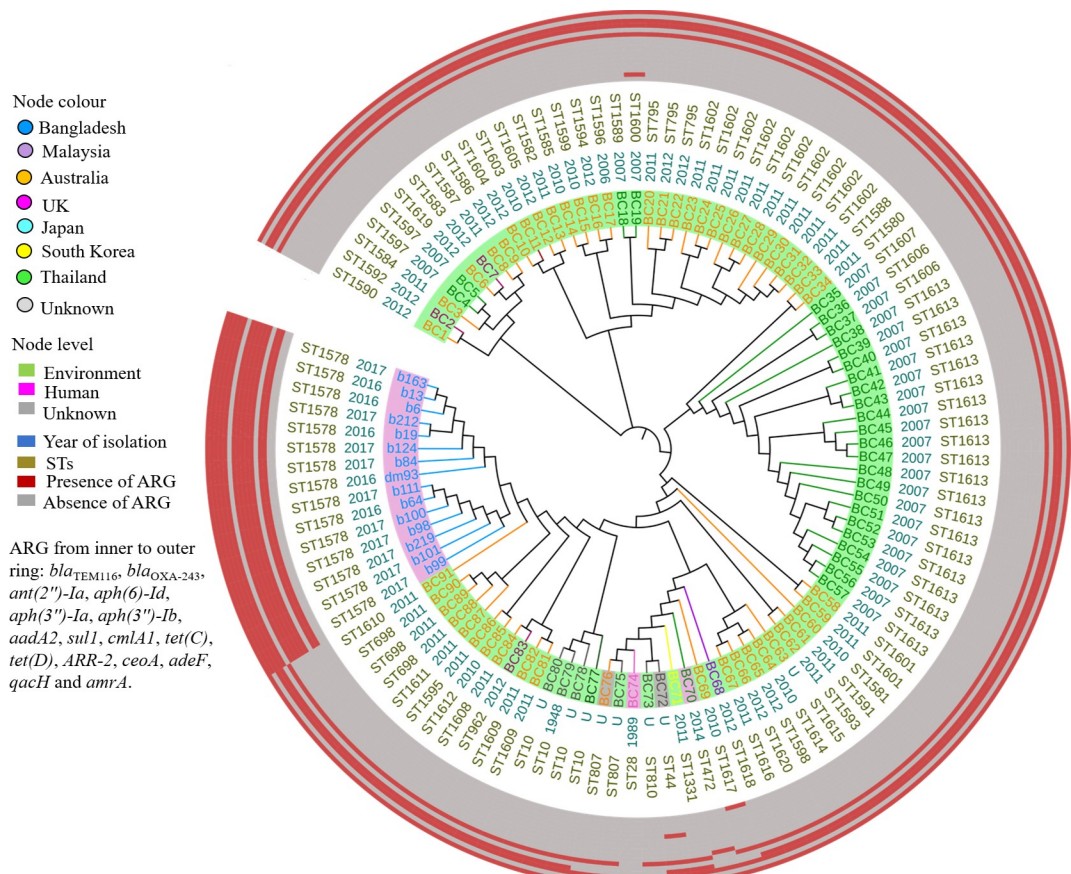

**Fig 4. A maximum likelihood tree of *B. cepacia* by core genome SNPs with epidemiological data and antimicrobial resistance genes.** Country of origin is represented by specific colour of node. Node level are highlighted according to source of sample. Global strains are stated with specific codes. Original strain IDs with corresponding codes are compiled with supplementary data (S1 Data). ARG, antimicrobial resistance genes; U, unknown.

breakpoints, 20% of *B. cepacia* (in this study) were resistant to ceftazidime and 93.33% intermediate resistant to levofloxacin. All the outbreak strains had high MIC value for amoxicillin-clavulanate, amikacin, gentamicin, fosfomycin, trimethoprim-sulfamethoxazole and colistin (Table 2). All were sensitive to meropenem (Table 2) [16]. MIC value of relevant antimicrobials are stated in Table 2. According to their MIC profiles, antibiotics deployed as empirical therapy to treat outbreak cases were invariably deemed to be inappropriate (Table 1). The antimicrobial resistance genes identified in the outbreak strains were identical (Fig 4; S1 Data). Likewise, *ampD* for all strains was homogenous, however, compared to *ampD* (BCAL3430) of *B. cenocepacia* strain J2315, we found 10 substitutions of amino acid in *ampD* (S1 Fig). The outbreak strains were shown to possess more resistant genes than other global strains (Fig 4; S1 Data).

The Bangladeshi outbreak strains shared some common virulence genes with the global strains (*bimA, boaAB, pilABCDNOQRSTV, gmhA, manC, wcbABCDEFGHIJKLMNOPQRST, wzm/wzt2, cheABDRWYZ, flgABCDEFGHIJKLMN, fliACDEFGHIJKLMNOPQRS, motAB, tsr, bspI2/bspI3, bspR2/bspR3/bspR4/bspR5, pmlI/bspI1, pmlR/bspR1, bapABC, basJ, bicAP, bipBCD, bopACE, bprABCDPQ, bsaKLMNOPQRSTUVXYZ, orgAB, spaP*, gene for T6SS), although two of the global strains (MSMB1338WGS.fsa_nt and DWS16B-4.fsa_nt) were negative for all genes from VFDB (Fig 5; S1 Data). There was variation in the presence of virulence

**Table 2. MIC value of *B. cepacia* in this study against relevant antibiotics (n = 15).**

| Strain ID | AUG[a] | Pip-Taz[a] | CRO[a] | CAZ[a] | CTX[a] | CEF[a] | IMP[a] | MEM[a] | CIP[a] | LEVO[a] | AK[a] | GEN[a] | Fos[a] | SXT-TRM[a] | Tige[a] | Cl[a] |
|---|---|---|---|---|---|---|---|---|---|---|---|---|---|---|---|---|
| dm93 | >256 | 4 | 8 | 4 | 16 | 8 | 8 | 4 | 2 | 4 | 64 | >256 | >256 | 32 | 16 | >256 |
| b19 | >256 | 8 | 16 | 8 | 32 | 16 | 8 | 4 | 1 | 4 | 256 | >256 | >256 | 64 | 4 | >256 |
| b06 | >256 | 8 | 16 | 8 | 32 | 16 | 8 | 4 | 1 | 4 | 256 | >256 | >256 | 32 | 16 | >256 |
| b13 | >256 | 8 | 16 | 8 | 32 | 16 | 8 | 4 | 2 | 4 | 256 | >256 | >256 | 32 | 16 | >256 |
| b64 | >256 | 2 | 4 | 2 | 8 | 8 | 8 | 2 | 2 | 4 | 64 | >256 | >256 | 64 | 8 | >256 |
| b84 | >256 | 8 | 16 | 4 | 16 | 16 | 8 | 4 | 2 | 4 | 256 | >256 | >256 | 64 | 16 | >256 |
| b98 | >256 | 2 | 4 | 2 | 8 | 8 | 8 | 2 | 2 | 4 | 64 | >256 | >256 | 64 | 8 | >256 |
| b99 | >256 | 16 | 32 | 8 | 32 | 32 | 8 | 4 | 2 | 4 | 256 | >256 | >256 | 32 | 4 | >256 |
| b100 | >256 | 8 | 16 | 4 | 32 | 16 | 8 | 4 | 2 | 4 | 256 | >256 | >256 | 64 | 8 | >256 |
| b101 | >256 | 2 | 4 | 2 | 8 | 8 | 8 | 2 | 2 | 4 | 64 | >256 | >256 | 32 | 8 | >256 |
| b111 | >256 | 2 | 4 | 2 | 8 | 8 | 8 | 2 | 1 | 2 | 256 | >256 | >256 | 64 | 4 | >256 |
| b124 | >256 | 2 | 16 | 8 | 16 | 16 | 8 | 4 | 2 | 4 | 256 | >256 | >256 | 64 | 16 | >256 |
| b163 | >256 | 16 | 32 | 16 | 32 | 8 | 8 | 4 | 2 | 4 | 128 | >256 | >256 | 32 | 16 | >256 |
| b212 | >256 | 8 | 16 | 16 | 16 | 16 | 8 | 4 | 2 | 4 | 128 | >256 | >256 | 64 | 16 | >256 |
| b219 | >256 | 16 | 32 | 32 | 32 | 32 | 8 | 4 | 2 | 4 | 64 | >256 | >256 | 64 | 16 | >256 |

[a]MIC values are indicated by mg/l

MIC values were determined in triplicate. AK, amikacin; AUG, amoxicillin-clavulanate; Pip-Taz, piperacillin-tazobactam; Cl, colistin; CRO, ceftriaxone; CAZ, ceftazidime; CTX, cefotaxime; CEF, cefepime; Fos, fosfomycin; GEN, gentamicin; IMP, imipenem; MEM, meropenem; CIP, ciprofloxacin; LEVO, levofloxacin; SXT-TRM, trimethoprim-sulfamethoxazole; Tige, tigecycline

genes between our clinical outbreak strains and previous clinical *B. cepacia* (LMG16656.fsa nt and LO6.fasta) isolated from CF patients (Fig 5; S1 Data).

## Discussion

Although *B. cepacia* is considered almost exclusively a pathogen for CF patients [3,7], our study reported *B. cepacia* bacteraemia predominantly from burns patients. *B. cepacia* are considered opportunistic human pathogens and can be transmitted to patients via environmental contamination or person-to-person contact [4–6,8,9]. Burns patients are generally more susceptible to infection due to impaired immune function. Risk factors for sepsis in burns include >20% of total body surface area (TBSA), inhalation injury, delayed burn wound excision, increased length of hospital stay, use of artificial medical devices and ICU admission [24–28]. Patients in burn ICUs are more vulnerable to septicaemia than general ICU patients [29]. Our findings demonstrate an outbreak of bacteraemia in DMCH caused by a single clone of *B. cepacia* ST1578, mostly confined to patients admitted in burn critical care units (Fig 2). The outbreak cases had a mean hospital stay of 43.9 days and straddled each other implying transmission via direct patient contact and/or patient to hospital environment and *vice versa*. Clinically, this outbreak was associated with a mortality rate of 31% (Fig 2; Table 1).

Previous studies show that burn sepsis accounts for 50–60% of deaths in burn patients [30,31]. In this study, mortality due to burn sepsis with bacteria other than *B. cepacia* (60.17%) was significantly higher than *B. cepacia* sepsis (33.33%) ($p<0.1$). A limitation of the current study was the lacked sufficient clinical information to access and to investigate why the mortality rate was significantly lower for *B. cepacia* cases. At the time of enrolment many of these patients were on "inadequate" antibiotics based on our susceptibility data. However, there are limitations to the interpretation of AST results for *B. cepacia* and we lack data on whether antibiotics were changed later during hospital stay. It is possible there were other factors favouring

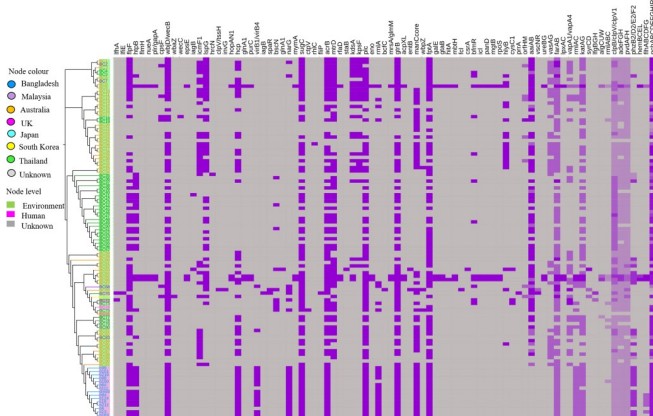

**Fig 5. Heatmap showing the presence of virulence genes in *B. cepacia*.** Global strains are stated with specific codes. Original strain IDs with corresponding codes are compiled with supplementary data (S1 Data). No VF was found in MSMB1338WGS.fsa_nt and DWS16B-4.fsa_nt. Apart from these two strains, other strains were shown to be positive for common virulence genes (*bimA, boaAB, pilABCDNOQRSTV, gmhA, manC, wcbABCDEFGHIJKLMNOPQRST, wzm/wzt2, cheABDRWYZ, flgABCDEFGHIJKLMN, fliACDEFGHIJKLMNOPQRS, motAB, tsr, bspI2/bspI3, bspR2/bspR3/bspR4/bspR5, pmlI/bspI1, pmlR/bspR1, bapABC, basJ, bicAP, bipBCD, bopACE, bprABCDPQ, bsaKLMNOPQRSTUVXYZ, orgAB, spaP*, gene for T6SS). These common VFs were not included in heatmap. Violet indicates the presence and grey indicates the absence of VF in the respective strain. Virulence genes in operon (*orfHM, aaiAB, vscNR, ureBG, vasAG, farAB, lpxAC, vapA1/vapA, rmlAC, katAG, syrCD, ifgBGH, algCUW, rmlABC, clpB.clpV/clpV1, adeFGH, pvdAFH, phzB2/D2/E2/F2, hemBCEL, flhABCDFG, pchABCDEFGHIR*) are represented by the intensity of violet—the increase in intensity is proportional to number of genes.

survival in the group infected with *B. cepacia*, including possibly relatively lower pathogenicity of *B. cepacia* as a cause of bacteraemia. *B. cepacia* has been described as a cause of pseudo-outbreaks [32]. Although this possibility should be considered here the timing of infections suggest this was not the case, and a persistent environmental source of cross contamination on the burn unit was a more likely cause. Unsurprisingly, radical infection control programs can mitigate the spread of infections and improve patient outcomes in burn units [33]. No IPC intervention was undertaken here because the outbreak was identified retrospectively.

MDR Gram-negative bacteria infections have become a serious challenge in health care settings as a result of both intrinsic and acquired resistance mechanisms, limiting therapeutic options [8]. *B. cepacia* is of concern due to their intrinsic resistance to clinically relevant antibiotics such as aminoglycosides and polymyxins [34]. In this study, all *B. cepacia* showed very high MIC breakpoints to amikacin and gentamicin mediated by resistance genes *ant*, *aph* and *aadA2* and/or overexpression of efflux pump such as ArmA, CoeA (Fig 4; S1 Data) [34–36]. *Burkholderia* spp. are typically resistance to colistin due to a unique intrinsic amino arabinose biosynthesis operon [33]. Genome sequencing also identified resistance genes such as *adeF*, *qacH*, *tetC* and *sul1* (Fig 4; S1 Data) which contribute to phenotypic resistance of *B. cepacia* (Table 2). Mutations in *ampD* are associated with the upregulation of ß-lactams degrading enzymes, PenB and AmpC. This mechanism has been found to be one of the causes of ß-lactam resistance in *B. cepacia* [18,34]. Mutations in *ampD* were identical in all *B. cepacia* analysed in our study and all the isolates had high MIC value for amoxicillin-clavulanate; however, MICs for piperacillin-tazobactam and cephalosporins were variable (Table 2). Perhaps enzymatic degradation was mainly responsible for putative ß-lactam resistance in this study although we did not evaluate the expression of ß-lactamase such as PenB, AmpC or PenA in relation to AmpD mutations [18,34]. Although the empirical antibiotics were shown

to be ineffective (Table 1), the relationship between patients' morbidity/mortality and antibiotics prescription cannot be fully explored as datasets are incomplete.

Virulence of *B. cepacia* is typically related to adhesins, invasins, intracellular pathogenicity, antiphagocytic factors, secretory and signalling systems [19,37]. A set of virulence genes in relation to all steps in pathogenesis were identified in Bangladeshi outbreak strains *B. cepacia* (Fig 5; S1 Data). However, whether other virulence determinants are associated with the ability to cause bacteraemia, and the role of patient factors, could not be determined by this current study.

*B. cepacia* identified in this study belonged to a novel clonal type ST1578 (Fig 3). The genetic background of the outbreak strains was very similar; however, a few variations were found regarding the presence of virulence genes (Fig 5; S1 Data). Compared to global strains from the database, resistance genes for aminoglycosides (*ant(2″)-Ia*, *aph(6)-Id*, *aph(3″)-Ib*, *aadA2*), fluoroquinolones (*qacH*), tetracycline (*tet(C)*) and sulphonamide (*sul1*) were found only among the strains isolated in this study (Fig 4; S1 Data). Virulence gene, *aai* was absent in our outbreak strain which was common in both human strains, LMG16656.fsa nt and LO6.fasta, (from *Burkholderia* Genome Database), but the overall virulence pattern of the outbreak strains was most similar to LMG16656.fsa nt (human strain isolated from the UK) than LO6.fasta (human strain isolated from Thailand). Compared to *B. cepacia* LO6.fasta, flagellar protein (lfg, lfh, lfi), secretory system (clpV1, vsc, iagB, spaR), regulatory protein (prrA) were absent in the Bangladeshi strains (Fig 5; S1 Data). Thus, it can be inferred that the Bangladeshi outbreak strains identified in this study are genetically distinct from global strains retrieved from the database.

Although we did not fully evaluate the epidemiological link between patients and the environment, epidemiological and molecular data suggest that the outbreak clone was circulating within DMCH over a protracted period (Fig 2). It is disconcerting that antibiotic pressure might have enhanced the elevation in antibiotic resistance during the outbreak [35,36]. Identification of environmental sources of the outbreak followed by patient management can reduce the risk of infections in vulnerable populations [4–6,10–13,29].

## Supporting information

**S1 Fig. Substitutions of amino acid in *ampD* compared to *ampD* (BCAL3430) of *B. cenocepacia* strain J2315.** Substitutions are underlined by red.
(TIF)

**S1 Data. Fig 4, Table 2, Fig 5.**
(XLSX)

## Acknowledgments

The Authors thank to Prof. Ismail Khan, Former Principal, Dhaka Medical College (DMC) for access to collecting clinical data and to Dr. Monira Pervin, Head, Department of Virology, DMC for providing laboratory support in Dhaka. We are grateful to Prof. S.M. Shamsuzzaman, Head, Department of Microbiology, DMC for providing support for blood culture in the microbiology department of DMC.

## Author Contributions

**Conceptualization:** Refath Farzana, Timothy R. Walsh.

**Data curation:** Refath Farzana, Md. Anisur Rahman, Kirsty Sands.

**Formal analysis:** Refath Farzana, Kirsty Sands.

**Funding acquisition:** Refath Farzana, Timothy R. Walsh.

**Investigation:** Refath Farzana, Md. Anisur Rahman, Kirsty Sands, Edward Portal, Ian Boostrom, Md. Abul Kalam, Brekhna Hasan, Afifah Khan, Timothy R. Walsh.

**Methodology:** Refath Farzana, Timothy R. Walsh.

**Project administration:** Refath Farzana, Md. Anisur Rahman.

**Resources:** Timothy R. Walsh.

**Software:** Refath Farzana, Kirsty Sands.

**Supervision:** Lim S. Jones, Timothy R. Walsh.

**Validation:** Refath Farzana, Lim S. Jones, Md. Anisur Rahman, Kirsty Sands, Edward Portal, Ian Boostrom, Brekhna Hasan, Afifah Khan, Timothy R. Walsh.

**Visualization:** Refath Farzana, Lim S. Jones, Timothy R. Walsh.

**Writing – original draft:** Refath Farzana, Lim S. Jones, Timothy R. Walsh.

**Writing – review & editing:** Refath Farzana, Lim S. Jones, Md. Anisur Rahman, Kirsty Sands, Edward Portal, Ian Boostrom, Md. Abul Kalam, Brekhna Hasan, Afifah Khan, Timothy R. Walsh.

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
