## [Decision Letter · Decision Letter 0]

16 Nov 2019

Dear Dr Refath Farzana:

Thank you very much for submitting your manuscript "Molecular and epidemiological analysis of a Burkholderia cepacia sepsis outbreak from a tertiary care hospital in Bangladesh" (#PNTD-D-19-01418) for review by PLOS Neglected Tropical Diseases. Your manuscript was fully evaluated at the editorial level and by independent peer reviewers. The reviewers appreciated the attention to an important problem, but raised some substantial concerns about the manuscript as it currently stands. These issues must be addressed before we would be willing to consider a revised version of your study. We cannot, of course, promise publication at that time.

We therefore ask you to modify the manuscript according to the review recommendations before we can consider your manuscript for acceptance. Your revisions should address the specific points made by each reviewer. 

When you are ready to resubmit, please be prepared to upload the following:

(1) A letter containing a detailed list of your responses to the review comments and a description of the changes you have made in the manuscript.

(2) Two versions of the manuscript: one with either highlights or tracked changes denoting where the text has been changed (uploaded as a "Revised Article with Changes Highlighted" file); the other a clean version (uploaded as the article file).

(3) If available, a striking still image (a new image if one is available or an existing one from within your manuscript). If your manuscript is accepted for publication, this image may be featured on our website. Images should ideally be high resolution, eye-catching, single panel images; where one is available, please use 'add file' at the time of resubmission and select 'striking image' as the file type. 

Please provide a short caption, including credits, uploaded as a separate "Other" file. If your image is from someone other than yourself, please ensure that the artist has read and agreed to the terms and conditions of the Creative Commons Attribution License at http://journals.plos.org/plosntds/s/content-license (NOTE: we cannot publish copyrighted images). 

(4) If applicable, we encourage you to add a list of accession numbers/ID numbers for genes and proteins mentioned in the text (these should be listed as a paragraph at the end of the manuscript). You can supply accession numbers for any database, so long as the database is publicly accessible and stable. Examples include LocusLink and SwissProt.

(5) To enhance the reproducibility of your results, we recommend that you deposit your laboratory protocols in protocols.io, where a protocol can be assigned its own identifier (DOI) such that it can be cited independently in the future. For instructions see http://journals.plos.org/plosntds/s/submission-guidelines#loc-methods

While revising your submission, please upload your figure files to the Preflight Analysis and Conversion Engine (PACE) digital diagnostic tool, https://pacev2.apexcovantage.com/ PACE helps ensure that figures meet PLOS requirements. To use PACE, you must first register as a user. Then, login and navigate to the UPLOAD tab, where you will find detailed instructions on how to use the tool. If you encounter any issues or have any questions when using PACE, please email us at figures@plos.org.

We hope to receive your revised manuscript by Jan 15 2020 11:59PM. If you anticipate any delay in its return, we ask that you let us know the expected resubmission date by replying to this email.

To submit a revision, go to https://www.editorialmanager.com/pntd/ and log in as an Author. You will see a menu item call Submission Needing Revision. You will find your submission record there. 

Sincerely,

Ruifu Yang

Deputy Editor

Alfredo Torres

Deputy Editor

Reviewer's Responses to Questions

**Key Review Criteria Required for Acceptance?**

**Methods**

-Are the objectives of the study clearly articulated with a clear testable hypothesis stated?

-Is the study design appropriate to address the stated objectives?

-Is the population clearly described and appropriate for the hypothesis being tested?

-Is the sample size sufficient to ensure adequate power to address the hypothesis being tested?

-Were correct statistical analysis used to support conclusions?

-Are there concerns about ethical or regulatory requirements being met?

Reviewer #1: (No Response)

Reviewer #2: This paper describes the use of whole genome sequencing to describe the phylogeny of B. cepacia isolates recovered over an 11 month period from in-patients hospitalised in burn critical care units in a large public hospital in Bangladesh. The experimental approaches used to examine the isolates are appropriate; however, the techniques used are not novel, although the application of WGS to B. cepacia isolates is still relatively uncommon. They are in the whole reasonably well presented although would benefit from clarification of some details (some suggestions below but not exhaustive) and provision of a rationale for the study should be provided.

Areas for improvement include clarification if it is indeed a retrospective study (if not then rationale for undertaking study-i.e. clinical suspicion, increase in incidence over baseline, etc.), if the 527 blood culture samples were all that were collected during the 11 month study period (and if these included the entire hospital’s workload or just those from the specific burn critical care units), if only 15 B. cepacia blood culture positive samples were identified during this period (again hospital wide versus specific units), if other B. cepacia isolates were recovered contemporaneously, including non-bloodstream infection related and environmental isolates. 

Minor points:

p5, ln10: Suggest re naming to “Culture, identification and antimicrobial susceptibility testing of B. cepacia isolates”

p5, ln108: include phrase re absence of EUCAST guidelines here rather than in Results section

p5, ln110: Suggest re write section to improve flow: i.e. provide detail re method of genomic DNA purification, include if manufacturer’s instructions were followed, usual nomenclature is 2 x 300 cycles, remove detail re actual GC content & depth of coverage to Results but include genome coverage calculation method, identification of resistance genes (as distinct from ‘retrieval’), if contigs were reordered against ATCC reference genome prior to annotation. 

P6, ln123 section: Clarify if MLSTs for the study isolates were determined from WGS reads or following traditional 7-locus amplification with Sanger sequencing.

P6, ln131 section: Describe process of generating reference genome assembly from dm93. Was a cgMLST approach considered for phylogenetic analysis and a wgMLST approach for virulence and resistance genotype analysis?

P6, ln134: change ‘aligemnt’ to ‘alignment’

No detail or references to statistical methods used are included.

Reviewer #3: This is an outbreak, however, it's not defined in the methodology as to what attempts were made to find out the source, if any.

The same needs to be clarified.

**Results**

-Does the analysis presented match the analysis plan?

-Are the results clearly and completely presented?

-Are the figures (Tables, Images) of sufficient quality for clarity?

Reviewer #1: (No Response)

Reviewer #2: The results describe the findings of an epidemiological investigation of 15 B. cepacia isolates recovered over an 11 period from in-patients hospitalised in burn critical care units in a large public hospital in Bangladesh. Clinical backdrop along with antimicrobial susceptibility and molecular characteristics of the isolates are presented. The authors use WGS data of 91 B. cepacia strains taken from a publically available genome database to demonstrate the distinct clonal nature of their isolates. It is evident from this manuscript that the authors have performed a level of background clinical investigation and sequence analysis of their isolates to contextualise and fully document the outbreak, although much more could be presented on WGS data analysis. Furthermore, the results are not presented clearly and as a consequence they are difficult to follow (some suggestions for improvement provided, but again not an exhaustive list). 

Minor points:

Include a section to detail WGS QC statistics, depth of coverage, %GC (from Methods), N50, etc. & also any statistics relating to the 91 global genomes if these differ from in-house isolate genomes. 

P9, ln197: Suggest this section be moved to before p9, ln178 section.

All of Table 1 is not visible, probably because of conversion to pdf

Fig 1: One blood culture positive case is not accounted for.

P7, ln144 section and Fig 2: It would be useful if some idea of bed capacity, occupancy rates and medical staff:patient ratios were available, as well as consideration of other underlying health conditions/risk factors (extent of burns, etc.) to better understand the mortality rate. Perhaps also an analysis of antibiotic treatment in relation to organism susceptibility could be considered and correlated to duration of infections and case outcome. These may well though form part of Table 1 and just be not visible to reviewer.

P9, ln178 section: Although this reviewer has little background in B. cepacia phylogeny, analysis of MLST and cgSNP is useful although in this instance the inclusion of only a few other human derived isolates does perhaps skew the results (possible that differences are distinct clade-associated), however perhaps WGS data from other human derived isolates are not available in publically accessible databases such as NCBI or BIGSdb. Could it be that two distinct clones exist within the Bangladeshi isolates (one recovered during the weeks around Feb 1st 2017), with evidence of microevolution within each over the course of the study? Were topologies of MLST and cgSNP trees compared? 

Fig 3: No scale is provided to assist with interpretation.

Fig 4: While very impressive looking, it is difficult to discern the node colour detail. No scale is provided to assist with interpretation. 

Table 2: The data could be better represented using S/R interpretations. 

P10, ln205 section and Table 2: Differences (greater than the tolerance of a doubling dilution on either side) in MIC values between strains are evident despite all isolates exhibiting the same resistance gene profiles; did the authors probe this further with respect to genotypes correlating with other antimicrobial resistance related mechanisms? 

P11, ln225: Improve details of differences in VF repertoire among Bangladeshi isolates; comment on potential significance of absence of lfiP, htpB, virB1/virB4 and phzB2/D2/E2/F2 from some Bangladeshi isolates; occurrence on plasmids, etc. 

P11, ln230: suggest rewording to “no virulence genes were identified in two of the global strains (MSMB1338WGS.fas_nt and DWS16B-4.fsa_nt) using VFDB.”

Fig 5: No scale is provided to assist with interpretation. Also, it is difficult to read the virulence genes

Reviewer #3: This is an outbreak, however, it's not defined in the methodology as to what attempts were made to find out the source, if any.

It should have been followed by the results of the attempts to find out the source.

It needs to be stated.

**Conclusions**

-Are the conclusions supported by the data presented?

-Are the limitations of analysis clearly described?

-Do the authors discuss how these data can be helpful to advance our understanding of the topic under study?

-Is public health relevance addressed?

Reviewer #1: (No Response)

Reviewer #2: The authors correctly concluded that the Bangladeshi B. cepacia isolates were from a single clone belonging to ST1578 and suggest that its continuance over the 11 month period was most likely related to transmission events with the burn critical care units. However, using the larger cgSNP dataset indicates that perhaps two similar clones (albeit of the same ST) were actually present; although without a scale on Figs 4 and 5 and also the inclusion of mainly environmental strains in the analysis with only a few other human-derived strains external to this time period and locale, it is difficult to be certain. 

Many important points are raised in the Discussion that unfortunately could not be adequately addressed due to lack of available information. In addition it is unclear as to how the conclusion relating to the “elevation of antibiotic resistance during the outbreak” was supported. Furthermore, many of the conclusions stated in the abstract, author summary and cover letter are not demonstrated in the Results and not developed in the Discussion.

Reviewer #3: The authors have concluded on the note of contamination, which hasn't been actually searched for in the whole manuscript.

It needs to be changed as the conceptualization of the manuscript isn't proper and it doesn't as such match with the title of the manuscript also.

**Editorial and Data Presentation Modifications?**

Reviewer #1: (No Response)

Reviewer #2: Some suggestions to improve the flow and readability of the text are included in the previous sections but these are not exhaustive.

Reviewer #3: Minor revision

**Summary and General Comments**

Reviewer #1: (No Response)

Reviewer #2: The manuscript entitled “Molecular and epidemiological analysis of a Burkholderia cepacia sepsis outbreak from a tertiary care hospital in Bangladesh” by Farzana et al details an investigation of 15 cases of B. cepacia bloodstream infection over an 11 month period in burns critical care units in Dhaka Medical College Hospital, Bangladesh. The authors examine if the cases are due to the spread of a single strain of B. cepacia. From using whole genome sequencing the authors establish that the 15 isolates were genetically very homogenous (compared to 91 other global B. cepacia, mainly from environmental sources) and harboured common antimicrobial resistance genes and virulence factors (distinct from the global B. cepacia) and conclude that the Bangladeshi isolates belonged to a single clone. A clone that was allowed to persist in the burn critical care units, due to inadequate antibiotic treatment and a lack of appropriate infection prevention and control measures. 

The topic of this manuscript is interesting considering the global struggle with multi-drug resistant organisms and will be of interest to the readers of the journal. It is one of the first papers that documents the use of whole genome sequencing to understand a clinical problem involving B. cepacia. It also highlights some of the challenges faced by the state health system in Bangladesh including overcrowding, inconsistent antimicrobial prescribing policies and variable infection control and prevention policies, all of which contributed to the unrestricted continuation of this strain over the 11-month study period. However, the manuscript would benefit from extensive revision and also a more in depth and appropriate analysis of the data with a better presentation and detailing of the results and discussion with appropriate references.

Reviewer #3: 1. In methods, it's not defined as to which breakpoints were used for MIC though described under Results in line 206 onwards, however, the year of CLSI Manual is not mentioned.

PLOS authors have the option to publish the peer review history of their article (what does this mean?). If published, this will include your full peer review and any attached files.

Reviewer #1: No

Reviewer #2: No

Reviewer #3: No

---

## [Decision Letter · Decision Letter 1]

13 Jan 2020

Dear Dr. Refath Farzana:

Thank you very much for submitting your manuscript "Molecular and epidemiological analysis of a Burkholderia cepacia sepsis outbreak from a tertiary care hospital in Bangladesh" (PNTD-D-19-01418R1) for review by PLOS Neglected Tropical Diseases. Your manuscript was fully evaluated at the editorial level and by independent peer reviewers. The reviewers appreciated the attention to an important topic but identified some aspects of the manuscript that should be improved.

We therefore ask you to modify the manuscript according to the review recommendations before we can consider your manuscript for acceptance. Your revisions should address the specific points made by each reviewer.

(1) A letter containing a detailed list of your responses to the review comments and a description of the changes you have made in the manuscript.

(2) Two versions of the manuscript: one with either highlights or tracked changes denoting where the text has been changed (uploaded as a "Revised Article with Changes Highlighted" file ); the other a clean version (uploaded as the article file).

(3) If available, a striking still image (a new image if one is available or an existing one from within your manuscript). If your manuscript is accepted for publication, this image may be featured on our website. Images should ideally be high resolution, eye-catching, single panel images; where one is available, please use 'add file' at the time of resubmission and select 'striking image' as the file type. 

Please provide a short caption, including credits, uploaded as a separate "Other" file. If your image is from someone other than yourself, please ensure that the artist has read and agreed to the terms and conditions of the Creative Commons Attribution License at http://journals.plos.org/plosntds/s/content-license (NOTE: we cannot publish copyrighted images). 

(4) Appropriate Figure Files 

Please remove all name and figure # text from your figure files upon submitting your revision. Please also take this time to check that your figures are of high resolution, which will improve both the editorial review process and help expedite your manuscript's publication should it be accepted. Please note that figures must have been originally created at 300dpi or higher. Do not manually increase the resolution of your files. For instructions on how to properly obtain high quality images, please review our Figure Guidelines, with examples at: http://journals.plos.org/plosntds/s/figures

While revising your submission, please upload your figure files to the Preflight Analysis and Conversion Engine (PACE) digital diagnostic tool, https://pacev2.apexcovantage.com/ PACE helps ensure that figures meet PLOS requirements. To use PACE, you must first register as a user. Then, login and navigate to the UPLOAD tab, where you will find detailed instructions on how to use the tool. If you encounter any issues or have any questions when using PACE, please email us at figures@plos.org.

We hope to receive your revised manuscript by Mar 13 2020 11:59PM. If you anticipate any delay in its return, we ask that you let us know the expected resubmission date by replying to this email.

To submit your revised files, please log in to https://www.editorialmanager.com/pntd/

Sincerely,

Ruifu Yang

Deputy Editor

Alfredo Torres

Deputy Editor

Reviewer's Responses to Questions

**Key Review Criteria Required for Acceptance?**

**Methods**

-Are the objectives of the study clearly articulated with a clear testable hypothesis stated?

-Is the study design appropriate to address the stated objectives?

-Is the population clearly described and appropriate for the hypothesis being tested?

-Is the sample size sufficient to ensure adequate power to address the hypothesis being tested?

-Were correct statistical analysis used to support conclusions?

-Are there concerns about ethical or regulatory requirements being met?

Reviewer #1: More detailed information may be need for the definition of the outbreak. Considering the DMCH is a tertiary care hospital, the Infection Control Division would have taken actions for better disinfection and clinical investigation, rather than observing the hospital stay over 11 months. Such a long time for a response to nosocomial infection emergency. Authors should provide more detailed information about clinical investigation.

Reviewer #3: Acceptable.

**Results**

-Does the analysis presented match the analysis plan?

-Are the results clearly and completely presented?

-Are the figures (Tables, Images) of sufficient quality for clarity?

Reviewer #1: Burkholderia cepacia sepsis outbreak among human is rare and neglected in clinic. The results basically match the analysis plan and the evidence is sufficient for the definition of the outbreak from the perspective of molecular epidemiological technology. The results are clearly demonstrated and the figures are of sufficient quality. If possibel, more background information of clinical investigation is better to predict the epidemiological links between cases. The gap between resistance gene prediction and clinical antimicrobial phenotype is often obvious.

Reviewer #3: -Are the results clearly and completely presented?

No. 

If we look at the table 1, an important finding is there that those patients who were given empirical antibiotic either levofloxacin or meropenem were discharged (though no. is small) and none of them died. Authors need to look at the susceptibility of these antibiotics in the given patient. Accordingly, this information may be included in the manuscript.

Another observation is that in case 1 & 3, despite on other antibiotics, patients get discharged. Did these patients have clinical signs of sepsis or do these cases fall under pseudobacteremia for which B. cepacia complex is well known?

**Conclusions**

-Are the conclusions supported by the data presented?

-Are the limitations of analysis clearly described?

-Do the authors discuss how these data can be helpful to advance our understanding of the topic under study?

-Is public health relevance addressed?

Reviewer #1: Generally, BC is not common pathogen in burn care unit. This is an out-of-control emergency of nosocomial infection. So donnot mention too much abouth the universality of BC infection in burn ICU, but take a deep research about the definition of the criminal bacteria and the links between cases.

Reviewer #3: -Are the conclusions supported by the data presented?

No.

Conclusions: Environmental contamination is a potential source of outbreak in a hospital setting with very poor infection control policies. 

The manuscript lacks the data to support this conclusion. Only the isolates from burn ward have been collected without even mentioning about the possible source. 

Despite pointing out earlier, it has still not been addressed as a limitation in the manuscript.

**Editorial and Data Presentation Modifications?**

Reviewer #1: Accept after minor revision in Investigating possible outbreaks in Methods section.

Reviewer #3: None

**Summary and General Comments**

Reviewer #1: This study is interesting for the Burkholderia spp. researchers and useful for the doctors in clinic in Burkholderia spp.-endemic regions. This paper is easily-understood, precise, and suitable for the aim of this maggazine, neglected diseases.

Reviewer #3: None

PLOS authors have the option to publish the peer review history of their article (what does this mean?). If published, this will include your full peer review and any attached files.

Reviewer #1: No

Reviewer #3: No

---

## [Editor Report · Decision Letter 2]

10 Feb 2020

Dear DR. Refath Farzana,

Thank you very much for submitting your manuscript "Molecular and epidemiological analysis of a Burkholderia cepacia sepsis outbreak from a tertiary care hospital in Bangladesh" for consideration at PLOS Neglected Tropical Diseases. As with all papers reviewed by the journal, your manuscript was reviewed by members of the editorial board and by several independent reviewers. The reviewers appreciated the attention to an important topic. Based on the reviews, we are likely to accept this manuscript for publication, providing that you modify the manuscript according to the review recommendations. 

Sincerely,

Ruifu Yang

Deputy Editor

Alfredo Torres

Deputy Editor
---

## [Editor Report · Decision Letter 3]

5 Mar 2020

Dear Dr. Farzana,

We are pleased to inform you that your manuscript 'Molecular and epidemiological analysis of a Burkholderia cepacia sepsis outbreak from a tertiary care hospital in Bangladesh' has been provisionally accepted for publication in PLOS Neglected Tropical Diseases.

Best regards,

Ruifu Yang

Deputy Editor

Alfredo Torres

Deputy Editor

---

## [Editor Report · Acceptance letter]

2 Apr 2020

Dear Dr. Farzana,

We are delighted to inform you that your manuscript, "Molecular and epidemiological analysis of a Burkholderia cepacia sepsis outbreak from a tertiary care hospital in Bangladesh," has been formally accepted for publication in PLOS Neglected Tropical Diseases.

Best regards,

Serap Aksoy

Editor-in-Chief

Shaden Kamhawi

Editor-in-Chief
